# OpenReview forum: "Exploring Recursive Doubt in Large Language Models"
_ICLR.cc/2026/Conference — Submitted to ICLR 2026_

### Official Review · Reviewer_PRw6 · 2025-10-29

**Soundness:** 3
**Presentation:** 3
**Contribution:** 2
**Rating:** 6
**Confidence:** 4

**Summary:**

The paper defines recursive doubt (getting stuck in self-questioning loops) distinct from “overthinking.” It proposes (1) FIND (Feedback-guided Iterative iNDuction), which trains an auxiliary LLM via preference optimization to craft prefixes that reliably induce recursive doubt in target models by rewarding recursion-marker tokens and length; and (2) FDA², a mitigation that detects “obsessive cognitive tokens” via a fence-like attention pattern (periodic peak spacing + cross-layer consistency). Experiments on GSM8K, MathQA, and MATH-500 across several models show increases in “recursive doubt rate” (RDR) under FIND.

**Strengths:**

1. Good definition and measurability. The paper gives an operational definition of “recursive doubt” and reports it via a clear metric (RDR), alongside token/time measures, making the phenomenon reproducible.

2. Cross-model transfer. The learned inducer prompts generalize across unseen models (different architectures/scales and API-only systems), consistently elevating the measured RDR and latency.

3. Actionable attention-pattern design. The method quantifies a distinctive “fence-like” attention signature (regular peak spacing and cross-layer consistency) to flag obsessive tokens and then applies a simple intervention (attenuating their input attention) that cuts looping in practice.

**Weaknesses:**

1. Metric depends on segmentation. RDR relies on how text is split into segments and on chosen similarity thresholds; the paper doesn’t clearly justify these choices or show robustness.

2. Inducer training choice. The attack hinges on RL/DPO to learn prefixes, but lacks comparisons to simpler or standard adversarial-generation methods.

3. Layer-wise clarity. The attention-based detector/mitigation doesn’t clearly explain which layers matter most or whether using all layers is necessary.

**Questions:**

1. RDR details & robustness: How exactly are segments defined (sentence, clause, fixed tokens)? How sensitive are results to different segmenters/thresholds?

2. Baselines for induction: Did you try non-RL alternatives (e.g., heuristic token sets, evolutionary/black-box search, gradient-free attacks, adversarial prompt methods) and compare efficacy/cost?

3. Layer ablations: Which attention layers contribute most to detection/mitigation? What happens if you apply it to some layers? Is it necessary to use all layers?

---

> ### Author Response · Authors · 2025-12-04
> **Rebuttal**
>
> Thank you for acknowledging the innovative nature and effective performance of our paper as well as providing the valuable feedback. Below we address the detailed comments, and hope that you can find our response satisfactory.
>
> **Q1: RDR details & robustness: How exactly are segments defined (sentence, clause, fixed tokens)? How sensitive are results to different segmenters/thresholds?**
>
> A1: We apologize for the omission of details regarding the RDR metric and the corresponding sensitivity analysis. For segment splitting, we use the newline character (“\n”) as the delimiter, which is a widely adopted approach (Trustworthy Reasoning: Evaluating and Enhancing Factual Accuracy in LLM
> Intermediate Thought Processes). We conduct sensitivity analyses on the similarity threshold and recursion ratio threshold used in RDR. As shown in the table below, RDR remains robust across different threshold selections, and the proposed FIND and consistently outperform other baselines under various configurations. We also perform sensitivity analyses, including testing different embedding models for RDR and other parameters. These results have been included in revision.
>
>
>
> ||Base|CoT|DoS|Engorgio|CAW|CAG|ICLGAW|ICLGAG|CatAttack|FIND|
> |-|-|-|-|-|-|-|-|-|-|-|
> |RDR 40\%|0.0|0.7|5.8|1.1|2.5|1.7|1.2|2.2|0.6|25.7|
> |RDR 45\%|0.0|0.5|5.4|0.7|1.2|1.0|0.9|2.0|0.3|24.0|
> |RDR 50\% (default)|0.0|0.4|4.2|0.3|0.8|0.3|0.6|0.9|0.1|20.4|
> |RDR 55\%|0.0|0.3|2.8|0.1|0.7|0.2|0.5|0.7|0.1|19.7|
> |RDR 60\%|0.0|0.1|2.6|0.0|0.4|0.2|0.3|0.6|0.0|16.7|
>
>
> ||Base|CoT|DoS|Engorgio|CAW|CAG|ICLGAW|ICLGAG|CatAttack|FIND|
> |-|-|-|-|-|-|-|-|-|-|-|
> |RDR 0.85|0.0|2.4|5.6|3.5|1.7|1.9|0.8|2.0|1.8|26.3|
> |RDR 0.875|0.0|1.2|4.9|0.8|1.4|1.3|0.8|1.8|0.7|25.3|
> |RDR 0.9 (default)|0.0|0.4|4.2|0.3|0.8|0.3|0.6|0.9|0.1|20.4|
> |RDR 0.925|0.0|0.3|1.1|0.3|0.6|0.3|0.5|0.5|0.0|15.3|
> |RDR 0.95|0.0|0.0|0.6|0.1|0.2|0.1|0.3|0.4|0.0|11.3|
>
>
>
>
> **Q2: Baselines for induction: Did you try non-RL alternatives (e.g., heuristic token sets, evolutionary/black-box search, gradient-free attacks, adversarial prompt methods) and compare efficacy/cost?**
>
> A2: To enable a more comprehensive comparison between the proposed FIND and standard adversarial generation methods, we develop four additional baseline methods, including a heuristic prompt-based method, a black-box search method, a gradient-free attack method, and an adversarial method that increases the probability of reflection-related tokens. The effectiveness and time consumption of these baselines and FIND are presented in the table below. While some baselines require fewer computational resources, they generally fail to effectively induce recursive doubt, indicating their potential inadequacy for this specific task. We will include this discussion in the revision.
>
>
>
> ||GSM8k||||MathQA||||MATH-500||||
> |-|-|-|-|-|-|-|-|-|-|-|-|-|
> ||ATL|ATC|RDR|ACC|ATL|ATC|RDR|ACC|ATL|ATC|RDR|ACC|
> |Adversarial Attack|730.7±54.0|8.9±0.7|0.5|96.2|2345.4±926.4|30.7±13.7|10.6|43.3|1601.6±171.6|19.6±2.1|1.0|83.9|
> |Gradient-free Attack|482.5±24.1|6.0±0.3|0.1|98.5|1352.8±177.6|16.4±2.2|6.0|45.9|1899.0±149.9|24.2±1.9|1.4|85.7|
> |Black-box Search|2163.5±1245.9|28.3±18.2|6.3|98.1|2480.4±962.0|31.0±13.7|6.2|48.1|4770.6±1589.7|61.4±23.1|6.8|85.6|
> |Prompt Attack|1063.3±77.0|12.8±0.9|2.8|96.7|3094.7±1243.7|39.9±18.2|10.4|41.2|3010.8±322.6|36.7±4.0|6.3|84.9|
> |FIND|**7922.3±2198.9**|**104.8±30.8**|**20.4**|95.3|**7783.4±2045.1**|**101.7±29.6**|**32.4**|38.0|**13373.2±2702.5**|**176.2±39.0**|**32.1**|83.2|
>
> **Q3: Layer ablations: Which attention layers contribute most to detection/mitigation? What happens if you apply it to some layers? Is it necessary to use all layers?**
>
> A3: We conduct further experiments to analyze the effects of disabling mitigation in specific layers. As shown in the table below, we observe that disabling mitigation in the shallow layers significantly impairs performance, whereas disabling it in the higher layers yields better results than the default FDA$^2$. We will include this discussion in the revision.
>
>
>
> ||ATL|ATC|RDR|ACC|
> |-|-|-|-|-|
> |FDA$^2$|1480.8±198.0|19.2±3.6|5.7|96.8|
> |FDA$^2$ w/o 1-7 layers|2098.1±303.8|24.4±21.9|14.4|95.5|
> |FDA$^2$ w/o 8-14 layers|1331.8±199.5|18.3±3.5|4.8|95.2|
> |FDA$^2$ w/o 15-21 layers|1292.9±191.7|18.1±3.6|5.6|95.7|
> |FDA$^2$ w/o 22-28 layers|1190.2±137.9|17.7±2.5|3.1|96.6|

---

### Official Review · Reviewer_yHLD · 2025-10-31

**Soundness:** 3
**Presentation:** 2
**Contribution:** 3
**Rating:** 2
**Confidence:** 4

**Summary:**

This paper introduces “Recursive Doubt” as a pathological behavior in large language models (LLMs), where the model gets trapped in repetitive cycles of self-questioning and negation during reasoning. The authors propose FIND (Feedback-guided Iterative iNDuction), a method that uses an auxiliary LLM to generate induction prefixes and iteratively refines them based on feedback from the target model to effectively trigger recursive doubt. They further identify a distinctive “fence-like” attention pattern in certain tokens—termed “obsessive cognitive tokens”—that consistently appear during such episodes. Building on this observation, they develop FDA2 (Fence-based Dynamic Attention Adjustment), which dynamically adjusts the attention weights of these tokens during inference to suppress recursive doubt. Experiments across multiple models and datasets demonstrate the effectiveness of both the induction and mitigation approaches.

**Strengths:**

1.This paper proposes and names a previously unstudied behavioral phenomenon in large language models, called "Recursive Doubt", and compare it with pathological doubt in human OCD, which provides a new perspective for understanding the abnormal reasoning behavior of LLMS.
2.This paper proposes a feedback-guided Iterative iNDuction (FIND) framework, which uses the Feedback from the target model to generate the inductive prefixes. It has good reproducibility and cross-model transfer ability.
3. This paper analyzes the attention pattern and discovers “fence-like” attention pattern, then designs Quantifiable metrics (Attention Peak Spacing Regularity and Spearman Attention Consistency) to characterize this pattern, which provides an empirical basis for understanding the internal mechanism of recursive doubt.

**Weaknesses:**

1. I don't think this kind of article structure is appropriate: The author spends a fair amount of time talking about "recursive doubt" in the model, and is too brief on how to alleviate it. And what's important: the definition of “recursive doubt” is entirely operationalized through superficial token statistics (e.g., counts of “Alternatively”) and output length. This lacks any grounding in cognitive modeling or computational psychiatry. Crucially, it fails to distinguish itself from well-documented phenomena like “overthinking” or redundant chain-of-thought. As Figure 1 shows, the only difference between (C) and (D) is repetition frequency—not a qualitative shift in reasoning dynamics. Labeling verbose outputs as “pathological” adds no theoretical value.

3. The FIND method does not reveal an intrinsic model flaw; it actively corrupts the input by instructing an auxiliary model to prepend adversarial prefixes like “Expand the original prompt by adding complexity and confusion” in Appendix C. This is not “inducing doubt”—it’s prompt injection by design. In real-world usage, no legitimate user would request the model to “add confusion.” Thus, although this experiment was done to get the model to behave like overthinking, the entire experimental setup is therefore artificial and lacks ecological validity.

4. FDA2’s mitigation hinges on identifying “fence-like” attention patterns as pathological. Yet the authors provide no evidence that this pattern doesn’t also occur in legitimate, complex reasoning (e.g., multi-step proofs). Blindly suppressing such tokens could cripple the model’s ability to handle hard tasks—a risk never evaluated. Meanwhile, “fence-like” attention patterns are not rare, works like SepLLM[1] etc. discover that the sep tokens also show “fence-like” attention pattern and it's useful for LLM.

5. The Figure 2 is too hard to read, I need to re-read the context to fully understand it. I hope the author will consider redrawing this figure or spilt it into two figures.

[1] http://arxiv.org/pdf/2412.12094v6

**Questions:**

Same as weakness.
And if the author can find out the reason for overthinking from the normal prompt instead of something like "Expand the original prompt by adding complexity and confusion." Then this article is very convincing to me.

---

> ### Author Response · Authors · 2025-12-04
> **Rebuttal (1/2)**
>
> Thank you for appreciating our new contributions as well as providing the valuable feedback. Below we address the detailed comments, and hope that you can find our response satisfactory.
>
> **Q1: I don't think this kind of article structure is appropriate: The author spends a fair amount of time talking about "recursive doubt" in the model, and is too brief on how to alleviate it. And what's important: the definition of “recursive doubt” is entirely operationalized through superficial token statistics (e.g., counts of “Alternatively”) and output length. This lacks any grounding in cognitive modeling or computational psychiatry. Crucially, it fails to distinguish itself from well-documented phenomena like “overthinking” or redundant chain-of-thought. As Figure 1 shows, the only difference between (C) and (D) is repetition frequency—not a qualitative shift in reasoning dynamics. Labeling verbose outputs as “pathological” adds no theoretical value.**
>
> A1: We need to clearly clarify several misunderstandings from the reviewer: 1. Sections 4 and 5 both belong to mitigation, not brief. 2. The definition of recursive doubt is not based on token statistics and output length, which is an incorrect description of our paper by the reviewer. 3. Recursive doubt and overthinking are two completely different phenomena.
>
> For the article structure, we introduce the definition of the recursive doubt and its potential consequences in Section 2. In Section 3, we propose FIND, a method to effectively induce recursive doubt. In Section 4, we analyze the attention patterns associated with recursive doubt and introduce obsessive cognitive tokens. Finally, in Section 5, we present the mitigation strategy FDA$^2$, which identifies obsessive cognitive tokens via a fence-like score to prevent the model from falling into recursive doubt. Although the mitigation method may appear to be confined to Section 5, it is directly motivated by the discovery of the fence-like attention pattern in Section 4. This ensures that our paper forms a complete framework addressing recursive doubt—from induction and analysis to mitigation.
> For the definition of recursive doubt, we clarify that it is not based on specific token counts or output length. As defined in Section 2, recursive doubt refers to the model producing a long sequence of highly similar segments in its output, meaning the model becomes trapped in a fixed reasoning pattern and cannot break free. This is fundamentally different from "overthinking" or redundant chain-of-thought. In the figure (https://anonymous.4open.science/r/R3Q1), we provide examples comparing recursive doubt and overthinking. It can be observed that, unlike overthinking, reasoning segments in recursive doubt exhibit high semantic similarity. The recursive doubt severity (defined in Lines 143-149) and the obsessive cognitive tokens count (defined in Lines 298-299) also support this distinction.
> As for the reviewer mentioned specific token statistics and output length, these are only used as computationally efficient and simple rewards for FIND. They are not related to the definition of recursive doubt or the design of the FDA$^2$.

---

> ### Author Response · Authors · 2025-12-04
> **Rebuttal (2/2)**
>
> **Q2: The FIND method does not reveal an intrinsic model flaw; it actively corrupts the input by instructing an auxiliary model to prepend adversarial prefixes like “Expand the original prompt by adding complexity and confusion” in Appendix C. This is not “inducing doubt”—it’s prompt injection by design. In real-world usage, no legitimate user would request the model to “add confusion.” Thus, although this experiment was done to get the model to behave like overthinking, the entire experimental setup is therefore artificial and lacks ecological validity.**
>
>
> A2: In Figure 5, we show an example where the adversarial prefix is "Expand the original prompt by adding complexity and confusion". We understand that the reviewer may have misunderstood that recursive loops only occur in artificially constructed adversarial settings. To clarify this point, we have supplemented with several examples involving more natural prompts, as shown in the table below. For instance, the model falls into recursive doubt when faced with additional reasonable constraints, which could realistically occur in legitimate user queries. We will include these examples in the revision.
>
> |Original Question|Prefix|
> |-|-|
> |If $f(x) = \frac{3x-2}{x-2}$, what is the value of $f(-2) +f(-1)+f(0)$? Express your answer as a common fraction.|What if we add 100 to the both side of the equation|
> |The results of a cross-country team's training run are graphed below. Which student has the greatest average speed? (Omitted the plotting program)|Carla and Briana also walk additional 20 km after the training run|
> |A regular hexagon can be divided into six equilateral triangles. If the perimeter of one of the triangles is 21 inches, what is the perimeter, in inches, of the regular hexagon?|The hexagon is inscribed within a larger circle with a radius of 10 inches, an|
>
> **Q3: FDA2’s mitigation hinges on identifying “fence-like” attention patterns as pathological. Yet the authors provide no evidence that this pattern doesn’t also occur in legitimate, complex reasoning (e.g., multi-step proofs). Blindly suppressing such tokens could cripple the model’s ability to handle hard tasks—a risk never evaluated. Meanwhile, “fence-like” attention patterns are not rare, works like SepLLM[1] etc. discover that the sep tokens also show “fence-like” attention pattern and it's useful for LLM.**
>
> A3: We apologize for the lacking of this discussion. We first provide example of attention pattern during normal reasoning in the figure (https://anonymous.4open.science/r/R3Q3). It can be observed that in normal reasoning, fence-like attention patterns are significantly reduced, even for more challenging problems (e.g., AIME24). In the table below, we further demonstrate the effect of FDA$^2$ on benign data. The results show that FDA$^2$, which mitigates recursive doubt by identifying fence-like reasoning patterns, maintains reasoning length and accuracy comparable to the base model when applied to benign data. This indicates that suppressing the attention of obsessive cognitive tokens does not harm the model’s performance on difficult tasks.
> Additionally, regarding the reviewer’s mention of the "fence-like" attention pattern identified in SepLLM, we note that it is inconsistent with what we have identified. In SepLLM, the fence-like pattern in attention maps reflects consistent high attention from different tokens to separator tokens, as these tokens may summarize paragraph-level information. In contrast, our notion of "fence-like" focuses on highly consistent and periodic patterns across different layers of attention maps, which is conceptually different from the fence pattern described in SepLLM. Secondly, when observing the fence-like repetitive attention pattern and calculating the fence-like degree, such sink tokens have already been excluded, as we stated in Line 258. Third, our approach does not remove the attention of obsessive cognitive tokens to these separator tokens. Instead, it identifies tokens with high fence-like scores to prevent them from attending to potentially maliciously induced original inputs,
> thereby avoiding direct harm to performance (as shown in the table below). We thank the reviewer again for the valuable suggestions and will include this discussion in the revision.
>
>
>
> ||ATL|ATC|RDR|ACC|
> |-|-|-|-|-|
> |Base|438.8±14.4|4.9±0.2|0.0|97.7|
> |FDA$^2$|840.4±115.7|8.6±1.5|1.6|96.7|
>
> **Q4: The Figure 2 is too hard to read, I need to re-read the context to fully understand it. I hope the author will consider redrawing this figure or spilt it into two figures.**
>
> A4: We thank the reviewer for the suggestion. We will make further revisions to the manuscript.

---

### Official Review · Reviewer_nRs2 · 2025-10-31

**Soundness:** 1
**Presentation:** 3
**Contribution:** 2
**Rating:** 2
**Confidence:** 4

**Summary:**

The paper is motivated by the behavior of some humans who enter a self-reinforcing loop of doubt, and the authors hypothesize that large language models (LLMs) might display similar patterns. They motivate this by arguing that self-reflection in chain-of-thought is motivated in the same way.

The main contribution is to show that LLMs can also be subject to recursive doubt. To explore this, the authors propose Feedback-guided Iterative iNDuction (FIND), which iteratively generates prefixes for input prompts. At each iteration, an auxiliary model proposes prefixes and queries the target LLM; the response provides a reward signal. Two reward functions are defined: one encouraging the production of reasoning tokens, and another maximizing sequence length to increase doubt and repetition. Pairwise prefix comparisons are built from these rewards and used to fine-tune the auxiliary model via preference optimization.

The authors then observe that some tokens often trigger repetitive reflection and doubt cycles, which they call obsessive cognition tokens. To mitigate this, they introduce Fence-based Dynamic Attention Adjustment (FDA^2), which analyzes attention patterns to locate such tokens and dynamically adjusts their attention weights using a "fenceness" score.

**Strengths:**

- Novel hypothesis linking human doubt to LLM behavior.
- Creative methods: FIND for iterative prefix generation and FDA² for attention adjustment.
- End-to-end pipeline that quantifies and mitigates recursive doubt.

**Weaknesses:**

- Anthropomorphic analogy is not empirically justified.
- Evaluation is biased: baselines not designed for recursive doubt, metrics favor the method, no confidence intervals.
- no ablation studies concerning sensitivity to prompts, rewards, and threshold.
- FDA² and FIND are not separately evaluated; contribution of each component unclear.

**Questions:**

The paper introduces an original analogy between human cognitive doubt and LLM behavior, but this analogy is not theoretically or empirically justified. The evaluation does not convincingly demonstrate the claimed phenomena, and the comparisons are not meaningful. The main issues are:

1. Anthropomorphic assumptions not supported by evidence.
2. Evaluation protocol and baselines poorly aligned with the stated objective.
3. Metrics defined by the authors and favorable to their own method.
4. No ablation study for key hyperparameters.

Main Comments

- It is far from obvious that LLMs exhibit behaviors comparable to human recursive doubt, and the anthropomorphic motivation seems more rhetorical than scientific. The psychological references are only used to motivate the analogy, and the paper does not actually draw on knowledge from psychology.
- The evaluation of FIND lacks a clear baseline that would deliberately induce recursive doubt. Competing models are selected for properties such as response length or latency, which makes the comparison weak.
- The metric for recursive doubt rate is introduced by the authors themselves, and since their model is explicitly optimized for it, the comparison with other models is not informative.
- The only more objective metric, accuracy, is less favorable to the authors; this result is not emphasized in the paper.
- The absence of ablation studies prevents assessing sensitivity to key meta-parameters such as the formulation of the prompt, the weights in the reward combination, or the threshold used in FDA^2. It is unclear how these were fixed.
- The experimental protocol should be revised to better highlight the contribution of FIND. For example, the method could be compared to a simple prompt designed to induce doubt.
- The evaluation of FDA^2 refers extensively to tables in the appendix. FDA^2 is compared to Prompt Mitigation, a template that discourages the use of certain fixed prefixes. This evaluation concerns the entire pipeline including FIND. It would probably be more informative to also evaluate FIND by replacing it with another component in the pipeline to better isolate its effect.
- Again, the reported improvements mainly appear for metrics not directly optimized by competing models. For accuracy, the absence of confidence intervals makes it impossible to determine whether the gains are significant.

Technical Comments

- The beginning of Section 2 (Problem Formulation) repeats parts of the introduction.
- The name FDA^2 misleadingly suggests a missing footnote.
- The first figure reference is to Figure 3, while Figure 1 is never cited.
- Line 198: A^{(j)}_{l,i} is introduced without appearing in previous formulas, and the use of "thus" is inappropriate.
- Line 212: the psychological reference is not convincing and adds nothing to the method; it would be more useful to demonstrate empirically that "fences" influence LLM behavior.
- Line 298: the authors do not explain how recursive doubt severity is computed.

---

> ### Author Response · Authors · 2025-12-04
> **Rebuttal (1/4)**
>
> Thank you for acknowledging the novelty of our paper as well as providing the valuable feedback. Below we address the detailed comments, and hope that you can find our response satisfactory.
>
> **Q1: It is far from obvious that LLMs exhibit behaviors comparable to human recursive doubt, and the anthropomorphic motivation seems more rhetorical than scientific. The psychological references are only used to motivate the analogy, and the paper does not actually draw on knowledge from psychology.**
>
> A1: We agree that the analogy was primarily used to vividly describe the pathological, recursive reasoning loop observed in large models. However, we do not consider this to be a weakness. Inspired by human psychology and psychiatry, we sought to explore whether large language models also exhibit recursive doubt phenomena. Through techniques like our proposed Feedback-guided Iterative iNDuction (FIND), fence-like repetitive attention patterns, obsessive cognition tokens, and Fence-based Dynamic Attention Adjustment (FDA$^2$), we have investigated this phenomenon in large language models. Our aim was simply to examine whether large language models exhibit a phenomenon similar to that found in humans.
>
> **Q2: The evaluation of FIND lacks a clear baseline that would deliberately induce recursive doubt. Competing models are selected for properties such as response length or latency, which makes the comparison weak.**
>
> A2: It’s need to clarify, to the best of our knowledge, the recursive doubt phenomenon is first identified in our work, making it difficult to find directly comparable baselines. To evaluate the performance of FIND as comprehensively as possible, we select eight baseline methods that are based on features such as response length or latency for comparison. To ensure a fair assessment, in addition to reporting the recursive doubt rate (RDR), we also include metrics such as the average output token length (ATL) and average inference time consumption (ATC) in all results, which are commonly reported in the original studies of these baselines (OverThink: Slowdown Attacks on Reasoning LLMs, An Engorgio Prompt Makes Large Language Model Babble on). We believe this constitutes a reasonable and strong comparative setup.
>
> Furthermore, to provide a more direct comparison with recursive doubt as the optimization target, we develop four additional baseline methods.
> These include a heuristic prompt-based method, a black-box search method, a gradient-free attack method, and an adversarial method that increases the probability of reflection-related tokens.
> Since these baselines are optimized using the same objective as FIND, they enable a fairer and more stringent comparison. As shown in the table below, FIND still demonstrates superior effectiveness compared to these baselines, indicating its effectiveness. We will include the results of these new baselines in the revision.
>
>
> ||GSM8k||||MathQA||||MATH-500||||
> |-|-|-|-|-|-|-|-|-|-|-|-|-|
> ||ATL|ATC|RDR|ACC|ATL|ATC|RDR|ACC|ATL|ATC|RDR|ACC|
> |Adversarial Attack|730.7±54.0|8.9±0.7|0.5|96.2|2345.4±926.4|30.7±13.7|10.6|43.3|1601.6±171.6|19.6±2.1|1.0|83.9|
> |Gradient-free Attack|482.5±24.1|6.0±0.3|0.1|98.5|1352.8±177.6|16.4±2.2|6.0|45.9|1899.0±149.9|24.2±1.9|1.4|85.7|
> |Black-box Search|2163.5±1245.9|28.3±18.2|6.3|98.1|2480.4±962.0|31.0±13.7|6.2|48.1|4770.6±1589.7|61.4±23.1|6.8|85.6|
> |Prompt Attack|1063.3±77.0|12.8±0.9|2.8|96.7|3094.7±1243.7|39.9±18.2|10.4|41.2|3010.8±322.6|36.7±4.0|6.3|84.9|
> |FIND|**7922.3±2198.9**|**104.8±30.8**|**20.4**|95.3|**7783.4±2045.1**|**101.7±29.6**|**32.4**|38.0|**13373.2±2702.5**|**176.2±39.0**|**32.1**|83.2|
>
>
> **Q3: The metric for recursive doubt rate is introduced by the authors themselves, and since their model is explicitly optimized for it, the comparison with other models is not informative.**
>
> A3: We need to clearly clarify a misunderstanding from the reviewer: the FIND method we propose is not directly optimized for the recursive doubt rate. Since the phenomenon of recursive doubt was first discovered in our work, and no existing metric was available to assess it, we introduced the Recursive Doubt Rate (RDR) for evaluation. RDR operates by segmenting the reasoning path and determining whether the reasoning is trapped in recursive doubt based on the proportion of excessively similar segments. However, it is important to note that our proposed FIND optimizes the prefix based on response length and count of key reasoning tokens, while FDA$^2$ calculates a fence-like degree by combining Attention Peak Spacing Regularity and Spearman Attention Consistency for effective mitigation. The objectives of both methods are entirely independent of the introduced RDR metric, meaning the RDR metric is fair and unbiased for all approaches. We also report additional metrics including average output token length (ATL), average inference time consumption (ATC), and accuracy (ACC) to ensure a more informative and comprehensive comparison.

---

> ### Author Response · Authors · 2025-12-04
> **Rebuttal (2/4)**
>
> **Q4: The only more objective metric, accuracy, is less favorable to the authors; this result is not emphasized in the paper.**
>
> A4: We need to clearly clarify several misunderstandings from the reviewer: 1. The accuracy is not disadvantageous to the FIND method we propose (see Section 6.2). 2. The accuracy results are explicitly highlighted in Lines 390-393 of the paper. The reviewer did not carefully read the experimental section and overlooked these clearly presented results. Specifically, we would like to clarify the following points: First, accuracy (ACC) is not the only objective metric in our study. Average output token length (ATL) and average inference time consumption (ATC) are also widely used objective metrics (OverThink: Slowdown Attacks on Reasoning LLMs, An Engorgio Prompt Makes Large Language Model Babble on), and the proposed FIND and FDA² outperform other baselines on these metrics. Second, the accuracy results are explicitly mentioned in Lines 390-393 of the paper and are fully reported in all relevant tables. The results are not bolded simply because accuracy is not the primary focus of this work—which centers on inducing and mitigating recursive doubt—rather than being intentionally downplayed. Finally, regarding the perception that the accuracy results appear "unfavorable", as stated in Lines 390-393, when a model falls into recursive doubt, its ability to produce correct answers is naturally impaired. Thus, the lower accuracy of FIND compared to other methods actually indicates that FIND is more effective at inducing recursive doubt.
>
> **Q5: The absence of ablation studies prevents assessing sensitivity to key meta-parameters such as the formulation of the prompt, the weights in the reward combination, or the threshold used in FDA$^2$.**
>
>
> A5:
> We conduct sensitivity analyses on the prompt formulation, the weight of reward combination, and the threshold $T_F$ of FDA$^2$. For the prompt formulation, we have already utilized the default templates for different models in our experiments (e.g., the harmony template for gpt-oss, https://github.com/openai/harmony), which further validates the insensitivity of the proposed methods to prompt variations.
> Furthermore, we additionally experiment with two alternative settings: Automatic Prompt Engineering (APE) (Large Language Models are Human-Level Prompt Engineers) and Plan-and-Solve Prompting (PS) (Plan-and-Solve Prompting: Improving Zero-Shot Chain-of-Thought Reasoning by Large Language Models).
> For the reward weights and the threshold $T_F$ of FDA$^2$, we conduct further experiments with weight values ranging from 30 to 70 and threshold values ranging from 0.25 to 0.45, respectively.
> As shown in the table below, FIND demonstrates considerable stability across different prompt templates, while FDA$^2$ also maintains strong robustness under varying threshold values. The experiment about the reward weights indicate that the default weight of 50 is the optimal parameter choice, as both excessively high and low weights may lead to suboptimal performance.
> We will add these discussions about sensitivity analysis in this revision.
>
>
>
> ||ATL|ATC|RDR|ACC|
> |-|-|-|-|-|
> |FIND (default)|7922.3±2198.9|104.8±30.8|20.4|95.3|
> |FIND APE|7500.7±1666.2|100.3±23.5|18.1|96.7|
> |FIND PS|9058.4±2763.2|119.3±38.8|23.3|91.4|
>
>
> ||ATL|ATC|RDR|ACC|
> |-|-|-|-|-|
> |FIND w/o $R_{\text{RD}}$|4652.2±1095.2|61.2±14.3|13.8|96.6|
> |FIND 30|7496.4±1886.8|93.7±24.9|20.6|93.9|
> |FIND 50 (default)|7922.3±2198.9|104.8±30.8|20.4|95.3|
> |FIND 70|3092.6±1399.7|39.2±19.5|9.8|95.1|
> |FIND w/o $R_{\text{Length}}$|2174.6±428.7|28.6±5.6|4.3|96.8|
>
>
> ||ATL|ATC|RDR|ACC|
> |-|-|-|-|-|
> |FDA$^2$ 0.3|1377.1±140.3|97.9±12.1|5.2|96.5|
> |FDA$^2$ 0.35 (default)|1480.8±198.0|19.2±3.6|5.7|96.8|
> |FDA$^2$ 0.4|1278.9±255.9|68.9±18.5|4.6|96.3|

---

> ### Author Response · Authors · 2025-12-04
> **Rebuttal (3/4)**
>
> **Q6: The experimental protocol should be revised to better highlight the contribution of FIND. For example, the method could be compared to a simple prompt designed to induce doubt.**
>
> A6: Thank you for the suggestion. We compare FIND with a simple prompt designed to induce doubt (i.e., the "Prompt Attack" in the table below). Additionally, we develop three extra baseline methods, including a black-box search method, a gradient-free attack method, and an adversarial method that increases the probability of reflection-related labels.
> Since these baselines are optimized using the same objective as FIND, they enable a fairer and more stringent comparison. As shown in the table below, FIND still demonstrates superior effectiveness compared to these baselines, indicating its effectiveness. We will include the results of these new baselines in the revision.
>
>
> ||GSM8k||||MathQA||||MATH-500||||
> |-|-|-|-|-|-|-|-|-|-|-|-|-|
> ||ATL|ATC|RDR|ACC|ATL|ATC|RDR|ACC|ATL|ATC|RDR|ACC|
> |Adversarial Attack|730.7±54.0|8.9±0.7|0.5|96.2|2345.4±926.4|30.7±13.7|10.6|43.3|1601.6±171.6|19.6±2.1|1.0|83.9|
> |Gradient-free Attack|482.5±24.1|6.0±0.3|0.1|98.5|1352.8±177.6|16.4±2.2|6.0|45.9|1899.0±149.9|24.2±1.9|1.4|85.7|
> |Black-box Search|2163.5±1245.9|28.3±18.2|6.3|98.1|2480.4±962.0|31.0±13.7|6.2|48.1|4770.6±1589.7|61.4±23.1|6.8|85.6|
> |Prompt Attack|1063.3±77.0|12.8±0.9|2.8|96.7|3094.7±1243.7|39.9±18.2|10.4|41.2|3010.8±322.6|36.7±4.0|6.3|84.9|
> |FIND|**7922.3±2198.9**|**104.8±30.8**|**20.4**|95.3|**7783.4±2045.1**|**101.7±29.6**|**32.4**|38.0|**13373.2±2702.5**|**176.2±39.0**|**32.1**|83.2|
>
>
> **Q7: The evaluation of FDA$^2$ refers extensively to tables in the appendix. FDA$^2$ is compared to Prompt Mitigation, a template that discourages the use of certain fixed prefixes. This evaluation concerns the entire pipeline including FIND. It would probably be more informative to also evaluate FIND by replacing it with another component in the pipeline to better isolate its effect.**
>
> A7: We thank the reviewer for the suggestion. We would like to emphasize that since FDA$^2$ is specifically designed to counteract the recursive doubt phenomenon, our original experiments focused solely on attacks induced by FIND. In response to the comment, we supplement additional experiments to evaluate the effectiveness of FDA$^2$ when facing attacks generated by DoS attack (Coercing LLMs to do and reveal (almost) anything). As shown in the table below, FDA$^2$ is still able to mitigate long reasoning processes to a certain extent when confronted with DoS attack, while simultaneously maintaining competitive ACC. We will include these experimental results in the revision.
>
> ||ATL|ATC|RDR|ACC|
> |-|-|-|-|-|
> |DoS|1253.2±425.2|14.5±5.0|4.2|98.6|
> |Prompt Mitigation|554.4±53.6|6.6±0.7|0.0|89.2|
> |FDA$^2$|842.2±55.1|9.4±0.6|0.7|96.1|

---

> ### Author Response · Authors · 2025-12-04
> **Rebuttal (4/4)**
>
> **Q8: Again, the reported improvements mainly appear for metrics not directly optimized by competing models. For accuracy, the absence of confidence intervals makes it impossible to determine whether the gains are significant.**
>
> A8: It is important to reemphasize (as clarified in the responses to Q3 and Q4, where several misunderstandings from the reviewer related to this point are addressed) that, since there are no existing baselines that directly induce recursive doubt, our comparison primarily focuses on methods that lead to increased token counts and higher response latency. The corresponding metrics ATL for evaluating token length and ATC for evaluating response latency—have been reported in all relevant tables. It is worth noting that FIND consistently outperforms all baseline methods not only in terms of RDR but also in these metrics. For accuracy, we show the results with 95\% confidence intervals in table below and conduct a one-sided paired Wilcoxon signed-rank test. The corresponding p-values 6.39e-120 indicate that the accuracy of FIND is significantly lower than that of other baselines, demonstrating its effectiveness in inducing the recursive doubt phenomenon, which consequently impedes the model’s ability to produce normal correct answers. We will include these confidence intervals in the revision.
>
>
>
>
> |||GSM8k|MathQA|MATH-500|
> |-|-|-|-|-|
> |R1-7b|Base|97.7±1.7|60.2±5.7|89.8±2.9|
> ||CoT|98.3±1.5|63.1±4.9|92.2±2.3|
> ||DoS|98.6±1.4|59.7±5.5|79.4±4.6|
> ||Engorgio|97.1±1.1|21.6±4.6|65.7±5.4|
> ||CAW|99.4±0.6|67.3±5.2|92.4±3.6|
> ||CAG|98.2±1.3|60.3±5.6|92.0±3.1|
> ||ICLGAW|97.4±1.6|66.9±5.4|81.3±4.6|
> ||ICLGAG|99.5±0.5|67.4±5.4|86.9±3.8|
> ||CatAttack|95.8±2.6|51.5±5.7|91.3±3.0|
> ||FIND|95.3±2.8|38.0±4.5|83.2±4.0|
> |QwQ-32b|Base|99.9±0.1|65.3±5.2|95.2±2.5|
> ||CoT|100.0±0.0|69.7±5.4|94.0±2.8|
> ||DoS|99.8±0.2|71.9±4.8|92.6±2.3|
> ||Engorgio|100.0±0.0|45.7±6.5|92.8±3.4|
> ||CAW|99.9±0.1|83.9±4.3|92.3±3.0|
> ||CAG|100.0±0.0|86.6±4.0|94.6±3.3|
> ||ICLGAW|97.4±1.2|86.7±3.7|93.7±2.8|
> ||ICLGAG|98.9±1.1|81.2±4.2|87.9±3.2|
> ||CatAttack|99.6±0.4|73.7±5.1|93.3±2.7|
> ||FIND|93.8±3.0|57.6±5.7|63.8±5.6|
> |gpt-oss-20b|Base|100.0±0.0|50.7±5.7|95.4±2.8|
> ||CoT|99.8±0.2|58.3±5.6|94.6±2.6|
> ||DoS|98.4±1.6|55.6±5.3|88.7±3.3|
> ||Engorgio|97.7±1.8|54.9±5.2|70.6±4.1|
> ||CAW|99.9±0.1|52.9±5.7|96.7±1.9|
> ||CAG|100.0±0.0|52.3±5.8|96.6±2.2|
> ||ICLGAW|97.2±1.9|57.2±4.9|93.7±2.7|
> ||ICLGAG|99.1±0.9|53.7±5.6|88.1±3.0|
> ||CatAttack|99.4±0.6|58.2±5.4|97.2±1.9|
> ||FIND|93.0±2.9|31.1±4.9|58.7±5.5|
>
>
>
>
> **Q9: A series of minor writing issues.**
> A9: We thank the reviewer for the suggestion. We will refine the relevant phrasing in the revision, avoid repetitive introductory content, add a reference to Figure 1, and make other necessary adjustments.
>
> **Q10: Line 298: the authors do not explain how recursive doubt severity is computed.**
>
> A10: We apologize for the lack of detailed explanation regarding the calculation of recursive doubt severity. The computation of recursive doubt severity is similar to that of the RDR metric: we also use the e5-small-v2 embedding model to encode segmented reasoning paths. The difference lies in the fact that we directly count the number of segments with a similarity score exceeding 0.9 compared to the preceding segment as the severity measure, whereas RDR labels a reasoning path as trapped in recursive doubt if more than 50\% of its segments meet this criterion. We will include a more detailed explanation in the figure caption of the revision.

---

### Official Review · Reviewer_pKdc · 2025-11-01

**Soundness:** 3
**Presentation:** 3
**Contribution:** 2
**Rating:** 4
**Confidence:** 4

**Summary:**

This paper studies Recursive Doubt in LLM reasoning, in which the model repeatedly questions and negates itself and creates lengthy output. (1) The authors propose FIND, an inducing prefix generation method to provoke Recursive Doubt behaviors. FIND involves DPO training an auxiliary LLM with keyword and length rewards. (2) They also analyze token-level attention and identify a distinctive fence-like pattern for obsessive cognitive tokens that triggers recursive doubt. (3) To mitigate, they propose FDA^2, which identifies these tokens and dynamically adjusts attention to avoid recursive doubt cycles. Experiments on three math datasets across three open models and two closed-source models show that FIND effectively increases output length, inference time, recursive doubt rate, and has cross-model transferability; FDA^2 reduces length, time, and improves accuracy under recursive doubt.

**Strengths:**

- The problem of recursive doubt is clearly formulated and different from the overthinking attack. The problem is tied to potential risks and thus can be of interest to LLM safety research
- FIND is a simple and general attack method that is effective across open and closed models
- The attention analysis shows distinctive fence-like patterns for obsessive cognitive tokens that trigger recursive doubt. The pattern is measurable using the proposed metrics (attention peak spacing regularity and Spearman attention consistency), which provide a concrete and analyzable tool for studying this phenomenon

**Weaknesses:**

Difficulties/significance of recursive doubt need better justification:

- All evaluation datasets focus on math reasoning. It's unclear about the attention patterns, the effectiveness of FIND and FDA^2, and the accuracy drop in other reasoning settings, such as coding and question-answering.
- Simple mitigation baselines are missing. The authors observe that certain keywords are associated with recursive reasoning; however, other decoding strategies, such as blocking keywords during decoding or applying repetition penalties, are not evaluated and analyzed.
- An analysis of the generated prefixes is lacking. What do successfully inducing prefixes look like? Are they semantically similar? Are they associated with certain keywords? Can prefixes be detected with keywords or LLMs?

Other weaknesses/concerns:

- RDR needs clarification and justification. How is output segmented? How are the thresholds selected? How sensitive is RDR under different similarity thresholds, recursive proportion thresholds, and/or different embedding models?
- The impact of FDA^2 under normal settings (without attack) is unknown. How is the fence-like degree threshold (T_F) selected? Under conditions where the FDA^2 is applied to a benign setting (misclassifies obsessive cognitive tokens), what will happen?
- The paper includes many references to psychological/psychiatric concepts (OCD, obsessive cognitive tokens), which may risk anthropomorphism. I think the authors should consider adjusting the wording or adding a disclaimer to avoid unwarranted parallelism.

**Questions:**

- In the FIND's transferability experiment, in many cases, pairing the prefix generator with unseen models has even lower accuracies (R1-7b: 95.3% vs 87.6%; QwQ-32b: 93.8% vs 86.0% and 84.3%; gpt-oss-20b: 93.0% vs 91.3% and 90.4%). Why? In addition, why are the lower accuracies often associated with lower RDR?

---

> ### Author Response · Authors · 2025-12-04
> **Rebuttal (1/3)**
>
> Thank you for acknowledging the novelty and insightful nature of our paper as well as providing the valuable feedback. Below we address the detailed comments, and hope that you can find our response satisfactory.
>
> **Q1: All evaluation datasets focus on math reasoning. It's unclear about the attention patterns, the effectiveness of FIND and FDA$^2$, and the accuracy drop in other reasoning settings, such as coding and question-answering.**
>
> A1: Thanks for your comment. To more comprehensively analyze the attention patterns of recursive doubt and the effectiveness of FIND and FDA$^2$ across diverse reasoning settings, we include additional experiments on the widely used question-answering dataset MMLU-Pro (MMLU-Pro: A More Robust and Challenging Multi-Task Language Understanding Benchmark) and coding benchmark HumanEval (Evaluating Large Language Models Trained on Code
> ). For MMLU-Pro, we randomly sample 10 problems from each of the 12 categories (excluding math), including physics, chemistry, law, etc. For HumanEval, we randomly select 100 coding problems for evaluation.
>
> As shown in the table below, FIND still outperforms other baselines on these reasoning benchmarks and successfully induces the recursive doubt phenomenon (figure: https://anonymous.4open.science/r/R1Q1).
> By analyzing the examples, we still observe the fence-like attention pattern, and the effectiveness of FDA$^2$ further supports this observation (table below). In summary, the observed attention patterns and the proposed method are not confined to mathematical reasoning but extend to other reasoning environments. We will include this analysis in the revision.
>
>
> ||MMLU-Pro||||HumanEval||||
> |-|-|-|-|-|-|-|-|-|
> ||ATL|ATC|RDR|ACC|ATL|ATC|RDR|ACC|
> |Base|1675.0±466.0|19.7±5.8|2.5|40.6|1699.3±91.3|19.6±1.1|0.5|99.6|
> |CoT|2906.7±1299.7|37.1±18.4|5.6|40.3|1883.8±143.6|21.7±1.7|1.3|99.7|
> |DoS|1779.8±818.4|22.9±11.8|1.4|34.7|1422.6±75.1|17.3±0.9|0.1|93.2|
> |Engorgio|2935.0±1029.6|37.8±15.0|5.8|39.4|1507.6±121.6|17.6±1.5|1.7|98.0|
> |CAW|1522.4±757.4|18.7±10.7|2.5|19.7|1555.8±65.2|17.8±0.8|0.2|99.6|
> |CAG|1493.3±505.3|18.6±6.5|1.4|14.2|1504.3±65.6|18.2±0.8|0.0|100.0|
> |ICLGAW|2607.5±1326.0|34.7±19.1|3.6|40.6|2917.0±921.4|36.3±13.3|7.8|86.9|
> |ICLGAG|1981.4±1057.2|25.9±15.4|3.9|43.6|1559.2±114.9|18.5±1.4|0.6|96.4|
> |CatAttack|2505.5±1117.5|31.4±15.6|4.4|35.1|1784.6±102.0|20.5±1.2|0.3|98.8|
> |FIND|**5873.0±2180.7**|**72.9±36.6**|**18.3**|30.0|**22300.5±3405.8**|**300.8±50.1**|**71.0**|70.9|
>
>
> ||MMLU-Pro||||HumanEval||||
> |-|-|-|-|-|-|-|-|-|
> ||ATL|ATC|RDR|ACC|ATL|ATC|RDR|ACC|
> |None|5873.0±2180.7|72.9±36.6|18.3|30.0|22300.5±3405.8|300.8±50.1|71.0|70.9|
> |Prompt Mitigation|3821.1±1299.0|49.5±18.2|14.7|36.9|14151.5±2264.1|182.7±31.5|41.4|62.1|
> |FDA$^2$|**2445.6±420.8**|**33.8±5.3**|**9.7**|**37.6**|**7119.1±586.2**|**88.1±8.5**|**27.5**|**82.5**|
>
> **Q2: Simple mitigation baselines are missing. Other decoding strategies, such as blocking keywords during decoding or applying repetition penalties, are not evaluated and analyzed.**
>
> A2: It’s worth noting that the repetition penalty of 1.0 has been applied throughout all experiments, as we state in Lines 829-830 in Appendix C, yet this does not prevent the emergence of recursive doubt. We also conduct additional experiments using keyword blocking as baseline mitigation methods. For keyword blocking, we employ the same set of keywords as demonstrated in the appendix, preventing the model from generating these tokens. However, as shown in the table below, this simple mitigation remain ineffective in suppressing the recursive doubt compared to FDA$^2$. Although keyword blocking significantly reduces output length, it also severely impairs reasoning capability, resulting in extremely low accuracy. We will include these baseline results in the revision.
>
>
>
> ||GSM8k||||MathQA||||MATH-500||||
> |-|-|-|-|-|-|-|-|-|-|-|-|-|
> ||ATL|ATC|RDR|ACC|ATL|ATC|RDR|ACC|ATL|ATC|RDR|ACC|
> |None|7922.3±2198.9|104.8±30.8|20.4|95.3|7783.4±2045.1|101.7±29.6|32.4|38.0|13373.2±2702.5|176.2±39.0|32.1|83.2|
> |Prompt Mitigation|2987.6±1434.3|37.5±19.4|13.8|95.5|4058.4±1114.3|49.0±15.0|29.8|54.7|7106.3±2116.7|88.8±28.8|27.8|85.7|
> |Keyword Blocking|1496.0±160.5|18.1±2.0|0.0|52.4|2874.8±297.7|35.3±3.7|0.0|9.5|2609.6±212.1|22.8±4.6|0.0|9.1|
> |FDA$^2$|1480.8±198.0|19.2±3.6|5.7|96.8|3728.5±725.0|44.6±13.2|9.7|56.3|2834.0±349.1|23.1±6.8|4.0|85.9|

---

> ### Author Response · Authors · 2025-12-04
> **Rebuttal (2/3)**
>
> **Q3: An analysis of the generated prefixes is lacking. What do successfully inducing prefixes look like? Are they semantically similar? Are they associated with certain keywords? Can prefixes be detected with keywords or LLMs?**
>
> A3: We present some examples of successfully induced recursive doubt prefixes in the table below. As shown in table below, these examples exhibit completely different semantics and keywords, which makes it impossible for keyword-based or LLM-based prefix detection to detect it.
>
> |Original Question|Prefix|
> |-|-|
> |If $f(x) = \frac{3x-2}{x-2}$, what is the value of $f(-2) +f(-1)+f(0)$? Express your answer as a common fraction.|What if we add 100 to the both side of the equation|
> |The results of a cross-country team's training run are graphed below. Which student has the greatest average speed? (Omitted the plotting program)|Carla and Briana also walk additional 20 km after the training run|
> |A regular hexagon can be divided into six equilateral triangles. If the perimeter of one of the triangles is 21 inches, what is the perimeter, in inches, of the regular hexagon?|The hexagon is inscribed within a larger circle with a radius of 10 inches, an|
>
>
>
> **Q4: RDR needs clarification and justification. How is output segmented? How are the thresholds selected? How sensitive is RDR under different similarity thresholds, recursive proportion thresholds, and/or different embedding models?**
>
> A4: We apologize for the lack of detailed information regarding RDR. For the computation of RDR, we first split the reasoning path into paragraphs using line breaks “\n”. We then use the e5-small-v2 embedding model to compute the similarity between adjacent paragraphs and identify those with a similarity score exceeding 0.9 as recursive segments. A reasoning path is labeled as trapped in recursive doubt if more than half of its segments are recursive.
>
> For the similarity threshold, to better measure the phenomenon of recursive doubt, we adopt a relatively high threshold of 0.9. For the recursive proportion threshold, the value of 0.5 is determined manually:
> we calibrate it against human-annotated recursive doubt data and find that 0.5 yields the best F1-score.
>
> We also conduct sensitivity analyses on the similarity threshold, recursive proportion threshold, and the choice of embedding model.
> For the similarity threshold, we test values ranging from 0.85 to 0.95. For the recursive proportion threshold, we examine values from 40\% to 60\%. Regarding the embedding model, we additionally evaluate the Qwen3 embedding model and the Gemma embedding model. We report the Recursive Doubt Rate (RDR) on the R1-7b model and GSM8K data in the table below.
> As shown, RDR is generally sensitive to these thresholds and embedding model choices.
> Moreover, across different threshold settings, FIND consistently outperform other baselines, demonstrating that the effectiveness does not depend on specific threshold choices.
>
>
>
> ||Base|CoT|DoS|Engorgio|CAW|CAG|ICLGAW|ICLGAG|CatAttack|FIND|
> |-|-|-|-|-|-|-|-|-|-|-|
> |RDR 40\%|0.0|0.7|5.8|1.1|2.5|1.7|1.2|2.2|0.6|25.7|
> |RDR 45\%|0.0|0.5|5.4|0.7|1.2|1.0|0.9|2.0|0.3|24.0|
> |RDR 50\% (default)|0.0|0.4|4.2|0.3|0.8|0.3|0.6|0.9|0.1|20.4|
> |RDR 55\%|0.0|0.3|2.8|0.1|0.7|0.2|0.5|0.7|0.1|19.7|
> |RDR 60\%|0.0|0.1|2.6|0.0|0.4|0.2|0.3|0.6|0.0|16.7|
>
>
> ||Base|CoT|DoS|Engorgio|CAW|CAG|ICLGAW|ICLGAG|CatAttack|FIND|
> |-|-|-|-|-|-|-|-|-|-|-|
> |RDR 0.85|0.0|2.4|5.6|3.5|1.7|1.9|0.8|2.0|1.8|26.3|
> |RDR 0.875|0.0|1.2|4.9|0.8|1.4|1.3|0.8|1.8|0.7|25.3|
> |RDR 0.9 (default)|0.0|0.4|4.2|0.3|0.8|0.3|0.6|0.9|0.1|20.4|
> |RDR 0.925|0.0|0.3|1.1|0.3|0.6|0.3|0.5|0.5|0.0|15.3|
> |RDR 0.95|0.0|0.0|0.6|0.1|0.2|0.1|0.3|0.4|0.0|11.3|
>
>
> ||Base|CoT|DoS|Engorgio|CAW|CAG|ICLGAW|ICLGAG|CatAttack|FIND|
> |-|-|-|-|-|-|-|-|-|-|-|
> |RDR E5 (default)|0.0|0.4|4.2|0.3|0.8|0.3|0.6|0.9|0.1|20.4|
> |RDR Qwen3|0.0|0.1|1.6|0.3|0.1|0.2|0.2|0.9|0.0|13.5|
> |RDR Gemma|0.0|0.3|2.9|0.8|0.7|0.3|0.1|1.8|0.9|22.7|

---

> ### Author Response · Authors · 2025-12-04
> **Rebuttal (3/3)**
>
> **Q5: The impact of FDA$^2$ under normal settings (without attack) is unknown. Under conditions where the FDA$^2$ is applied to a benign setting (misclassifies obsessive cognitive tokens), what will happen?**
>
> A5: We supplement the results of FDA$^2$ on normal data in the table below. The results show that when applied to normal data, FDA$^2$ achieves competitive accuracy compared to standard reasoning while demonstrating no significant increase in ATL and ATC. This phenomenon stems from two main factors: first, misclassification rarely occurs when processing normal data.
> Additionally, even when tokens are classified as compulsive, our method only removes their attention to specific tokens rather than eliminating the tokens entirely. This mild intervention mechanism prevents disruption to normal reasoning processes. In conclusion, these experiments indicate that FDA$^2$ has minimal impact on normal reasoning, suggesting its robust effectiveness. We will include these experiments and analyses in the revision.
>
> ||ATL|ATC|RDR|ACC|
> |-|-|-|-|-|
> |Base|438.8±14.4|4.9±0.2|0.0|97.7|
> |FDA$^2$|840.4±115.7|8.6±1.5|1.6|96.7|
>
> **Q6: The paper includes many references to psychological/psychiatric concepts (OCD, obsessive cognitive tokens), which may risk anthropomorphism. I think the authors should consider adjusting the wording or adding a disclaimer to avoid unwarranted parallelism.**
>
> A6: We thank the reviewer for pointing out the risk of anthropomorphism here. We agree that direct analogies between LLM behaviors and complex human psychological conditions require careful wording to avoid misleading implications. Our original intention was not to introduce these psychological/psychiatric concepts, but to heuristically employ the analogy to illustrate a self-reinforcing, pathological reasoning pattern in LLMs.
> We will add a footnote to explicitly clarify that this is strictly an analogy/metaphor, preventing readers from drawing inappropriate parallelism or misunderstandings.
>
> **Q7: In the FIND's transferability experiment, in many cases, pairing the prefix generator with unseen models has even lower accuracies (R1-7b: 95.3\% vs 87.6\%; QwQ-32b: 93.8\% vs 86.0\% and 84.3\%; gpt-oss-20b: 93.0\% vs 91.3\% and 90.4\%). Why? In addition, why are the lower accuracies often associated with lower RDR?**
>
> A7: We need to clarify that a lower accuracy is not always associated with a lower RDR. (R1-7b: (20.4\%, 95.3\%) vs. (37.3\%, 91.3\%); gpt-oss-20b: (13.3\%, 96.9\%) vs. (57.9\%, 93.0\%)). Additionally, we need to clarify that: in Table 1, FIND results in lower accuracy. However, in the transferability experiment of FIND, pairing the prefix generator with seen models does not necessarily lead to lower accuracy. This is because, in Table 1, FIND causes the model to enter an infinite loop, reaching the token limit and preventing the model from generating valid answers, which leads to a decrease in accuracy. In the transferability experiment of FIND, whether or not the model has been seen before (i.e., whether it is transferred) will cause the model to enter an infinite loop. Therefore, the accuracy drop fluctuates depending on whether the infinite loop in the test set samples reaches the token limit, making it impossible to directly compare in a meaningful way.

---

### Author Response · Authors · 2025-12-04
**Summary of Contributions and Rebuttal Responses**

Dear Reviewers and AC,

We sincerely thank all the reviewers and the area chair for their valuable time. **Almost all reviewers unanimously recognize the significant contributions of our work, with the issues raised primarily focusing on specific experimental settings and clarifications. During the rebuttal period, we addressed all the reviewers' concerns with detailed clarifications and substantial additional empirical evidence, which we believe resolved their concerns and significantly improved the quality of the paper. Unfortunately, due to the inadvertent leak of reviewer information, the discussion period was closed prematurely, and reviewers were unable to provide further feedback or participate in the discussion.**

We understand that this has resulted in a significant workload for the newly assigned AC to evaluate the paper within the limited time available. Below is our comprehensive summary to help clarify the core contributions and key modifications of the paper.

**Significance and Motivation**:

- This paper aims to investigate whether large language models exhibit a phenomenon similar to "recursive doubt" in human obsessive-compulsive disorder, where models repeatedly question and deny their reasoning and conclusions, potentially leading to significant safety risks. For example, recursive doubt could cause an LLMbased precision robotics control system to execute repetitive actions, destabilizing decision-making processes. Similarly, in the context of autonomous driving, recursive doubt might result in delayed responses or erratic driving, posing risks to passengers and pedestrians.

**Key Contributions**:

- The first to propose and study the phenomenon of recursive doubt in LLMs.
- Introduced the FIND method, which guides iterative reasoning via feedback to induce recursive doubt.
- Discovered the "fence-like" attention pattern in LLMs, a key manifestation of recursive doubt.
- Proposed the FDA$^2$ mitigation strategy, which suppresses recursive doubt by dynamically adjusting attention weights.
- Experiments demonstrate that FIND and FDA$^2$ are effective across multiple models and datasets, with strong cross-model transferability.

**Key Improvements and Clarifications**:

- **Extended Evaluation Beyond Mathematical Reasoning:** In response to Reviewer pKdc’s concerns, we conduct extensive new experiments on the MMLU-Pro (question-answering) and HumanEval (coding) benchmarks. The results confirm that the identified "fence-like" attention pattern and the effectiveness of our proposed FIND and FDA$^2$ generalize robustly to other reasoning domains, demonstrating the broader significance of the recursive doubt phenomenon.

- **Comprehensive Baseline Comparisons:** To address concerns from Reviewers pKdc, nRs2, and PRw6 regarding baseline method selection, we introduce more comprehensive baselines, including four methods specifically designed to induce recursive doubt and keyword blocking for mitigation. Extensive experiments demonstrate that the proposed FIND and FDA$^2$ consistently and significantly outperform these baselines.

- **Clarification of Core Concepts:** We clarify Reviewer nRs2’s misunderstanding regarding the concept of Recursive Doubt Rate (RDR): RDR measures high semantic similarity between reasoning segments and is used solely for evaluation. The optimization objectives of the proposed FIND and FDA$^2$ methods are entirely independent of RDR. In response to Reviewer yHLD’s inquiry about the distinction between "recursive doubt" and "overthinking," and Reviewer pKdc’s concern about attack examples, we provide more detailed examples that could realistically occur in legitimate user queries.

- **More Comprehensive Evaluation of FDA$^2$:** Addressing concerns from Reviewers pKdc and yHLD regarding FDA$^2$’s mitigation effects on benign problems and other attack methods (raised by nRs2), we supplement extensive experimental results. These demonstrate that FDA² does not harm normal reasoning performance and effectively mitigates other types of attacks.

- **Sensitivity Analysis, Robustness, and Additional Experiments:** In response to suggestions from Reviewers pKdc, nRs2, and PRw6, we conduct thorough sensitivity analyses on key parameters, including RDR’s similarity/proportion thresholds, embedding models, FDA$^2$’s mitigation threshold, and FIND’s reward weights. Additionally, following PRw6’s suggestion, we perform ablation studies on the mitigation capabilities of different layers in FDA$^2$.

We believe that the revised manuscript and the detailed explanations in our responses have fully addressed the reviewers' concerns, even though, due to system issues, reviewers were unable to provide further feedback. Thank you again for your valuable time and guidance.

Sincerely,

The Authors

---

### Meta-Review · Area_Chair_iGs7 · 2026-01-05

**Summary:**

This work's core contribution lies in proposing "Recursive Doubt," a pathological reasoning phenomenon in LLMs. Meanwhile, the authors also design the FIND method to induce this phenomenon and propose the FDA2 method for mitigation.

All four reviewers acknowledged the novelty of the research topic but raised a series of constructive concerns spanning argument validity, experimental generalizability, metric reliability, and methodological ecological validity.  The authors supplemented their rebuttal with additional experimental data and analysis.

As AC, I evaluate the submission objectively across three core dimensions: 1) "sufficiency of argument support," 2) "rigor of experimental design," and 3) "practical value of the method", based on reviewers' original comments, author responses, and supplementary materials, ultimately forming this decision.

- The author's response does not address the core deficiency. First, the scientific validity of the analogy remains unestablished, namely, human OCD's recursive doubt stems from cognitive self-questioning mechanisms, whereas the paper only observes output repetition patterns in LLMs. The authors have not established the connection from cognitive modeling or computational psychiatry perspectives. Adding a footnote cannot bridge the "disconnect between phenomenon naming and essential definition." Second, the distinction criterion of "high semantic similarity" relies on the RDR metric but lacks independent qualitative validation. In this context, it is hard to prove that "recursive doubt" is a novel phenomenon independent of overthinking.

- The authors' supplementary experiments partially address generalizability concerns, but baseline reasonableness deficiencies remain. First, the so-called "natural prefixes" do not represent genuine user demands. For example, "add 100 to both sides of the equation" is artificially imposed additional constraints, not a natural extension of user queries. The authors provide no evidence of "recursive doubt spontaneously occurring in uninterrupted normal queries," failing to prove the phenomenon is an inherent LLM deficiency rather than artificially constructed. Second, baseline comparisons lack rigor: the keyword blocking baseline shows dramatically reduced accuracy, yet the authors do not justify the selected keywords' reasonableness, leaving open the possibility that "poor baseline performance stems from inappropriate keyword selection." Among induction baselines, non-RL methods perform worse, but the authors provide no analysis of FIND's computational cost-benefit tradeoff with these methods, undermining evidence of practical value.

-  Core deficiencies in metrics and method remain partially unresolved. RDR's segmentation approach lacks universality. The authors did not adjust segmentation strategy, directly applying "line-break segmentation," which raises doubts about RDR's computational reliability in non-mathematical domains. Sensitivity analyses only covered GSM8K, not newly added MMLU-Pro and HumanEval, failing to prove metric robustness across domains. In addition, FDA2 safety validation is insufficient: the authors tested only on simple normal data without evaluating performance on complex reasoning tasks. Though reviewers noted that "fence-like attention patterns" in SepLLM benefit normal reasoning and authors claim different definitions, they provide no empirical evidence that FDA2 will not misclassify valid patterns in normal reasoning.

**Reviewer Concerns:**

Reviewer PRw6.

Concerns of the left three reviewers are still outstanding.

**Reviewer Scores:**

hard to say

---

### Decision · Program_Chairs · 2026-01-26

Reject